# Advance Care Planning and Goals of Care Discussion: Barriers from the Perspective of Medical Residents

**DOI:** 10.3390/ijerph20043239

**Published:** 2023-02-13

**Authors:** Laiane Moraes Dias, Mayra de Almeida Frutig, Mirella Rebello Bezerra, Williams Fernandes Barra, Luísa Castro, Francisca Rego

**Affiliations:** 1Faculty of Medicine, University of Porto, 4200-319 Porto, Portugal; 2João de Barros Barreto University Hospital, Federal University of Pará, Belém 66075-110, PA, Brazil; 3Institute of Cancer of São Paulo-ICESP, São Paulo 01246-000, SP, Brazil; 4IMIP, Instituto de Medicina Integral Professor Fernando Figueira, Recife 50070-902, PE, Brazil

**Keywords:** communication skills, oncology, medical residency, advance care planning

## Abstract

Background: Advance care planning (ACP) and goals of care discussion involve the exploration of what is most important to a person to prepare for health-care decision making. Despite their well-established benefits, they are still not frequently performed in clinical oncology practice. This study aims to describe the barriers to discussion goals of care with oncology patients from the perspective of medical residents. Methods. This cross-sectional and qualitative study applied the “Decide-Oncology” questionnaire, adapted to Portuguese language, to assess barriers to goals of care discussion among medical residents from three university hospitals in Brazil. Residents were asked to rank the importance of various barriers to discuss goals of care (ranging from 1—extremely unimportant to 7—extremely important). Results: Twenty-nine residents answered the questionnaire (30.9%). The most reported barriers were related to patients and their families’ difficulty in understanding and accepting the diagnosis and the prognosis as well as patients’ desire to receive full active treatment. Furthermore, the physician and external factors such as lack of training and lack of time to have these conversations were also very important barriers. The identification of the key barriers that limit the discussion of ACP and early palliative care referrals can certainly help to prioritize the next steps for future studies aimed at improving ACP and goals of care discussions.

## 1. Introduction

Advance care planning (ACP) and goals of care discussions involve the exploration of a person’s values, beliefs, and what is most important to each person: to ensure concordance between the clinical care received by a patient and the care he or she has wished for. The context for each differs, however, as ACP conversations focus on preparing for future health-care decisions, whereas goals of care discussions focus on current health-care decisions. Thus, goals of care discussions are also an important part of the advance care planning process [1,2].

One of the principles of good care is respect and receptivity to patients’ wishes and values; thus, it is important to understand patients’ perspectives about cancer treatment [3]. High-intensity life support is often provided even when patients or their families may prefer to focus on comfort care in spite of evidence suggesting that use of unwanted technology is associated with negative outcomes, such as decreased quality of life and low satisfaction with end-of-life care [4]. Robust evidence shows that ACP is associated with a wide variety of benefits, such as: less moral distress by health care professionals, higher rates of advanced directives (AD) registrations, reduced hospitalization, intensive and futile treatment at the end-of-life, greater probability of the patient dying in the chosen place, greater satisfaction with the quality of care, and less risk of stress, anxiety and depression in family members during bereavement [1,5]. All in all, it improves the consistency of care with patients’ values and goals in various patient populations, including oncology patients [6].

Notwithstanding the importance of ACP, many health-care providers are still reluctant to engage patients in this discussion. Some barriers include concerns about causing distress for a person or decreasing hope for life-sustaining outcomes, personal discomfort with death and dying, lack of experience in discussing end-of-life issues and limited education and training in ACP [1]. By the same token, implementing interventions to improve communication and decision making about goals of care and ACP in oncology settings requires an understanding of the perspectives of patients, families and clinicians. Likewise, awareness of barriers enables the development of tailored interventions that are more likely to improve professional practice and training programs [7,8]. Considering the scarcity of studies about the difficulties of medical residents on discussing ACP and about their training on this topic, this study aims to determine, from the perspective of medical residents, the barriers to the discussion goals of care and ACP with oncology patients.

## 2. Aim

This study aims to explore barriers to goals of care and advance care planning discussion in patients with cancer from the perspective of Brazilian medical residents and to compare such barriers according to years of medical residency.

## 3. Methods

### 3.1. Type and Questionnaire Study

To identify and evaluate the barriers that impact on discussing goals of care and on advance care planning of oncology patients, according to the perception of resident physicians, a cross-sectional quantitative and qualitative study was developed through the application of questionnaire “Decide Oncology” [4].

Statements about doctors’ difficulties focused on three pillars: (1) barriers on the discussion about goals of care; (2) barriers on the approach of interrupting active cancer therapies (for example: chemotherapy, radiation therapy); and (3) barriers of referring to palliative care teams. These difficulties were further stratified into barriers related to patients, doctors and/or related to the health system or external factors. It is a Likert questionnaire where each session has statements about the difficulties in discussing the advance care planning, classified from 1 to 7 points, where 1 is the extremely unimportant difficulty; 2, very unimportant; 3, little unimportant; 4, neither important nor unimportant; 5, slightly important; 6, very important; and 7, extremely important [4].

The questionnaire also included questions about sociodemographic characteristics (age, gender, ethnicity, religion, year of graduation and medical fellowship and specialty) and questions about the perception of the importance and the degree of formal training on ACP and discussing goals of care.

The questionnaire’s adaptation and translation to Portuguese was conducted. The direct and reverse translation from English into Portuguese (Brazil) by four translators was completed followed by the evaluation of six specialists in palliative care, who were invited as judges to analyze the clarity of the language, content validity and the translation’s adequacy. For that, the Content Validity Coefficient (CVC) was used, as suggested by Hernández-Nieto (2002), and the criterion of CVC > 0.80 was used to consider whether the items have been semantically adjusted [9].

The questionnaire was adapted into an electronic version (Google forms) and sent via email to medical residents from three Brazilian reference oncology centers of three university hospitals (Belém-Pará, Recife and São Paulo).

This study was submitted and approved by the Ethics Committee of João de Barros Barreto University Hospital (No 4.376.747). All subjects had to give their informed consent to participate in this study.

### 3.2. Study Population

Resident physicians who had training in oncology or palliative care units (e.g., oncology, internal medicine, geriatrics and hematology) from three university hospitals were contacted. To assess possible differences between residents according to the time since graduation, they were divided into two groups: one with up to two years of residency and the other with three years or more. Out of ninety-four resident physicians referred, 29 responded.

### 3.3. Statistical Analysis

The electronic surveys were pooled and compiled using descriptive statistics, including median and interquartile interval (1st Q; 3rd Q) for quantitative variables and counts and proportions for categorical variables. Survey responses were presented by median scores. The distribution of scores was compared between groups of residence years using Mann–Whitney’s non-parametric test. Data analysis was performed using SPSS^®^ Statistics (version 26.0; SPSS Inc., Chicago, IL, USA). In all tests, values of *p* < 0.05 were considered significant.

### 3.4. Qualitative Analysis

A content analysis based on Bardin [10] was conducted, and two categories, namely intrinsic and extrinsic factors, were developed based on the content found in the answers to the two open questions: 1—“Reflecting on the barriers which you rated as very important or extremely important in Section 1, what specific suggestions do you have about ways to overcome these barriers and make it easier for health-care providers to talk with patients and their family members about goals of care?” 2—“What is currently working well to promote communication and decision making about goals of care between health-care providers and patients and their family members?”.

As these questions were related to the unique issue “Suggestions to improve communication and decision making about goals of care”, we condensed the answers according to the content analysis. A system of categories was developed to analyze the contents following three phases: pre-analysis (contact with the material to be analyzed, detailed reading of the content); exploration of the material (choice of coding units, classification, grouping of words by meaning); and treatment of results—inference and interpretation (identification of the latent content, the meaning behind the apprehended content) [10]. This process was conducted by two authors independently, and they then reached a consensus about the categories.

## 4. Results

### 4.1. Quantitative Analysis

#### 4.1.1. Participants

The survey was distributed to a total of 94 medical residents (10 paper, 84 electronic forms) between December 2020 and March 2021. A response rate of 30.9% (100% answered the paper questionnaire and 22.6% the electronic version) was achieved. Respondent demographics are outlined in Table 1.

#### 4.1.2. Barriers to Discuss ACP and Goals of Care

Most of the residents described the patient and family barriers as the most important, such as “patients’ difficulty in understanding their diagnostic and on accepting their prognosis”, “patients’ desire to receive full active treatment”, “the lack of an advance directive”, “family members’ difficulty in understanding their loved one’s prognosis”. As for the physician’s barriers, “lack of training” was the most important barrier described. “Lack of time to have conversation” and “lack of availability of substitute decision maker(s)” were two of the barriers described as very important. The importance given to the “lack of time to have conversations about goals of care” was significantly different between the two groups of years of residency (*p* = 0.034), being a more important barrier among those with more years of residency (median scores of 4 and 6 for groups of up to 2 years and more than 2 years of residency, respectively; see Table 2).

#### 4.1.3. Barriers to Discuss the Discontinuation of Cancer-Directed Therapies

The barriers with higher scores were the patient and family barriers, which were also most frequently described to the discontinuation of cancer-directed therapies. Additionally, patients’ expectations regarding the benefits of therapy was another barrier, as were patients’ and families’ poor appreciation of prognosis or denial of likely survival duration and patients’ inflated expectation of the benefits from further cancer-directed therapy. The most described physicians’ barriers were the discordance with other specialists in estimating prognosis/length of survival, which was followed by difficulty on estimating patient prognosis/length of survival, uncertainty of the benefits of further active cancer therapy, and patient age (whereby they had more difficulty suspending active treatment in younger patients). External barriers were ranked lower than patient/family and physician-level barriers (Figure 1B). Therefore, comparing the two groups of residents, “patients’ expectations regarding the benefits of therapy” and “patient age” were the barriers that presented significant different scores and higher importance in the group with more than two years of residency (*p* < 0.001 and *p* = 0.025, respectively, Table 2).

#### 4.1.4. Barriers to Early Palliative Care Referrals

The external barriers such as lack of access to palliative care services in the community and in the hospitals, lack of multidisciplinary team (social workers, nurse practitioners, etc.) to aid in patient support/referral process to palliative care, and the patient or family refusal of referral, were the barriers most described to early palliative care referrals (Figure 1).

#### 4.1.5. The Willingness to Discuss ACP and Goals of Care

Most of the respondents were very or extremely willing to initiate (82.8%) and lead (86.2%) ACP and goals of care discussions.

Physician’s perception about the ability to discuss ACP and goals of care and the quality of training on communication varied.

Most of the residents’ self-reported having average skill in conducting goals of care discussions with patients and families (44.8%), while 89.6% ranked their priority for learning these skills as high (as 4 or 5 out of 5). By contrast, 93.1% had formal training regarding communication and goals of care, and 77.7% considered their quality of training from moderate to extremely high (Table 3).

### 4.2. Qualitative Analysis

A total of 27 participants (93%) answered the open-ended question about the strategies and suggestions to improve decision making about goals of care. The answers to the question “What could facilitate decision making about goals of care and advance planning care in clinical practice?” enables identifying two categories: factors related to the physician (intrinsic factors) and factors related to the health system (extrinsic factors) (Table 4).

The intrinsic factors—related to the physicians—most described to improve decision making about goals of care were: communication skills training (P8—“Training about breaking bad news, palliative care and death during graduation and medical residency. Little is taught about it”), which was followed by a good physician–patient relationship (P16—“Establishing a good doctor–patient–family relationship from the beginning of follow-up and treatment can facilitate conversation about difficult issues”); and the capacity to talk about the prognostic earlier in the course of the disease (P6—“Clearly expose the patient’s prognosis as soon as possible so that the patient is able to participate more actively in decision making”). Empathetic communication was also mentioned (P18—“having good communication skills with sincerity and clarity, and know how to welcome the patient during the conversation, without taking away their hope”) (Table 4).

Extrinsic factors such as enabling access to palliative care and interdisciplinary teams (P26—“Constant training of the medical and multidisciplinary team as well as regular meetings with a multidisciplinary and palliative care team, also involving patients and family members”), as well as having enough time to discuss goals of care and advance care planning during consultation (P12—“Good relationship with patient and family and to have time for patient and family education”) (Table 4).

## 5. Discussion

The frequent barriers to the discussion of goals of care and ACP present in the literature included lack of time, discomfort with difficult discussions, fear of affecting patients’ hope and emotional coping and lack of training in communication strategies about end-of-life [5,11,12,13]. In the present study, the “Decide-Oncology” questionnaire was used to evaluate the barriers on discussing goals of care. It classifies the barriers between those of physicians, those of patients and families, and those of external barriers. Regarding the difficulties to discuss goals of care, most were related to patients’ and family members’ factors, such as the difficulty for patients to understand and accept their prognosis. These findings corroborate those of the Decide-study [7] and Decide-Oncology study [4], in which health-care professionals on general medical wards and physicians in oncology ambulatory identified patient and family-related factors as the most important barriers to goals of care discussion. Some external factors such as the lack of time for these discussions was also considered a significant barrier in the present study, which is in agreement with the aforementioned authors [4,7].

However, the physicians’ perception about the patient’s difficulty to understand and accept his/her prognosis may reflect, at least in part, on the doctors’ difficulty in predicting prognosis—a well-reported barrier in this study—and even with the well-established prognosis; they may have difficulties on communicating this prognosis in an objective and clear way [4,7]. It is important to consider that this study involved exclusively medical residents—who spend at least one month at oncology and/or palliative care services—unlike the Decide-Oncology study [4], which included mostly oncologists, possibly with more experience than medical residents.

When comparing groups of residents by years of medical residency, 3rd and 4th year residents (generally on oncology, geriatric, hematology or palliative medicine residency) considered having more difficulties to discuss about goals of care related to the patients’ expectations regarding the benefits of therapy, and difficulties with the lack of time to have this conversation. In the field of oncology, communication skills are paramount, as the life-threatening or life-limiting nature of the conditions encountered in this field create multiple challenges for practitioners both those in training and seasoned oncologists. The ability to communicate about a new diagnosis, poor prognosis, relapse, imminent death, or death itself is not innate, nor is it commonly taught at any level of training [14]. According to Perron et al. (2015), the clinical experience alone is not enough; such skills should be taught and trained [15]. It could reflect the needs of continuous education in medical communication skills at the postgraduate level.

Physicians who deal with patients with serious illnesses need to have great communication skills. Clarifying cancer treatment goals and aligning expectations are essential to engage patients in timely and meaningful ACP15. It requires good illness understanding and realistic expectations about prognosis so that patients can express their values and make decisions in a timely manner [16]. For example, data from the Cancer Care Outcomes Research and Surveillance (CanCORS) study showed that 69% of patients with stage IV lung and 81% of patients with stage IV colorectal cancers expected that chemotherapy would be curative [17,18]. In contrast, patients who understood that the role of chemotherapy was to reduce disease burden and control symptoms were more likely referred to appropriate hospice services and receive less aggressive care at the end-of-life [19]. Efforts to optimize communication are essential to effectively address gaps in illness understanding and moderate expectations for benefit while preserving hope, empathy, and the therapeutic relationship between patients and their health-care team [20].

Granek (2013) explored oncologists’ views on end-of-life discussions and revealed various physician-related barriers to ACP, including personal discomfort with death and dying, reluctance to communicate painful information, perception of other physicians’ failure, focusing too much on a cure or treatment, along with a lack of experience with breaking bad news, especially in end-of-life care and lack of mentorship on training end-of-life communication strategies [13]. Regarding the discontinuation of directed cancer therapies, this study also found the discordance with other specialists in estimating prognosis/length of survival, which was followed by difficulty on estimating patient prognosis and uncertainty of the benefits of further active cancer therapy as important barriers. It is known that earlier follow-up by a palliative care team could improve the communication with patients with incurable disease in helping them to prepare for the end of life through shared decision making among patients, their families, and health-care providers [21].

There is evidence that the use of chemotherapy near the end of life is not related to its likelihood of providing benefit. So, the suspension of aggressive treatment before the end of life is an indicator of good quality care, especially in the last two weeks of life [22]. This indicator is often difficult to capture but reflects the notion that optimal palliative care in the final weeks of life generally includes withdrawal of cancer-aggressive and directed therapies, including withholding aggressive resuscitation measures [23]. This concept may be more difficult for patients to comprehend, as the mistaken notion of ongoing cancer treatment is often connected to maintaining good care and hope [4]. In the present study, medical residents indicated lack of access to palliative care services in the community and at hospitals, and a lack of multidisciplinary teams (social workers, nurse practitioners, etc.) hindered patient support or referral process to palliative care, as well as the patient or family refusal of referral to palliative care, as opposed to the Decide-Oncology study that found patient and family desire to maintain hope as a significant barrier. It is worth mentioning that palliative care is not yet well integrated into public health in all regions of Brazil, unlike developed countries, where palliative care is well inserted and structured at different levels of health care.

Detering et al. [2] identified trained facilitators, patient-centered discussions and the systematic education of doctors, among other elements, as crucial to successful advance care planning. They also found that patients welcome ACP and expect health professionals to initiate this kind of discussion. Another recent systematic review found that healthcare professionals’ training in ACP had positive effects on their knowledge, attitude and skills about communication in end-of-life care [24]. The use of decision aids and advanced technology, such as audio-visual material and online tutorials; workshops using role play, training content focused on ACP communication skills and the needs and experience of patients in decision-making processes, contribute to ACP training programs effectiveness [24,25,26]. There are few studies about advance care planning skills designed for medical residents that have been evaluated for their effectiveness [27,28,29]. Some authors demonstrated that communication skills training programs may improve residents’ skills in breaking bad news with a positive change in their emotional reactions after training and minimizing insecurity to talk about challenging issues such as prognosis [24,27,28,29,30]. In the qualitative analysis of the present study, most of the residents suggested the communication skills training and the ability to talk about prognosis early as strategies to facilitate advance care planning and goals of care discussion. Most of them self-reported an average skill level to discuss goals of care with their patients and also ranked it as a high priority to learn those skills, although most of them already had a formal training regarding communication and goals of care discussion. It is probably because most of the residents are already at the third or fourth year of residency, in oncology–hematology or palliative care services at university hospitals, and deal with patients with advanced diseases frequently in their clinical practice.

Different tools may be effective for clinicians with varying expertise and competency in end-of-life communication skills. Myers et al. [1] demonstrated in a systematic review the instruments that could facilitate the ACP process, such as Respecting Choices [31], a program with trained facilitators, and the Physician Orders for Life Sustaining Treatment (POLST) that is a standardized form containing medical orders that reflect a patient’s treatment preferences [32]. They demonstrated that exposure to these tools, and others, if compared to controls, could result in an increased advance directive completion; increased appointment of a surrogate; more involvement in end-of-life decisions; better consistency between patient wishes and medical interventions undertaken at end-of-life [1,33,34]. Most studies still present data from the point of view of healthcare providers, and more research from the patients’ perspective is still needed [1,24]. Barriers related to patients and family members were the most reported in the present study as well as in previous studies involving only health-care professionals [4,7].

In Brazil, a competency framework of palliative medicine for geriatricians’ residents was developed to guide the teaching of essential and desired skills of palliative care in geriatrics settings [35]. It includes communication’s skills such as: to explore the elderly patient and family’s understanding of illnesses, concerns, goals and values and planning treatment that fits these priorities (advance care planning); to perform an effective patient-centered communication on breaking bad news or delivering prognostic information, in addition to adequately communicating to the patient on how to perform advance directives, based on bioethical and legal principles [35]. In other areas such as oncology, a competency framework would certainly help to standardize the skills and competencies of resident physicians. In our study, a great majority of residents described continuing education in palliative care and communication skills training as suggestions for improving shared decision-making about goals of care.

Given the lack of studies that address barriers to discussing goals of care, especially in the Brazilian reality, evaluating the difficulties from the perspectives of medical residents and their relative importance to discussing this topic are the strengths of this study. This study is the first step for the adaptation and validation of the Decide-Oncology questionnaire [4] to the Brazilian–Portuguese language. Nonetheless, some limitations included a small sample, because it involved only medical residents from three university hospitals; due to this, it is possible that our findings may not be representative of all eligible respondents. Furthermore, a positive response bias among those who did participate is possible with potential for higher response rates among those most engaged in palliative and end-of-life care.

## 6. Conclusions

This study described the medical residents’ difficulties in discussing goals of care and ACP with oncology patients. The most important reported barriers were related to patients and their families, such as their difficulty in understanding and accepting the diagnostic and the prognosis. The physician and external factors such as lack of training and lack of time to have these conversations were equally described as very important barriers. In addition, the importance and willingness to initiate and lead ACP and goals of care discussions were demonstrated by most residents in this study, who consequently indicated the need for training in this area as a priority. Not to mention that there are various interventions to prepare patients and families for ACP and end-of-life communications, strategies to overcome barriers for health-care providers and practical tips to initiate these difficult conversations [1,8,24,34]. Moreover, the identification of the key barriers that limit the discussion of ACP and early palliative care referrals could help to prioritize the next steps for future studies aimed at improving advanced care planning and goals of care discussions [7]. All things considered, communication skills training and tools that enhance medical residents and clinicians’ ability to listen with empathy, to strengthen doctor–patient relationships and to clearly discuss prognosis, will help clinicians better support patients and families through shared decision making based on the patient’s values and experiences.

## Figures and Tables

**Figure 1 ijerph-20-03239-f001:**
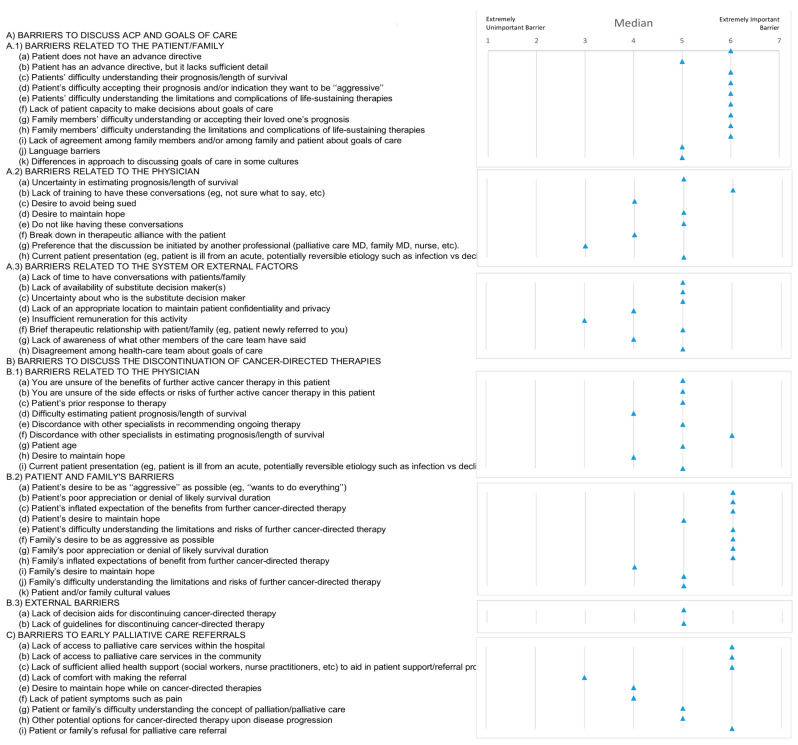
Barriers of patient, physician, and external factors to initiating goals of care discussions (A), to interrupt cancer-directed therapies (B), and to refer to palliative care (C) (median scores).

**Table 1 ijerph-20-03239-t001:** Socio-demographic profile of the 29 medical residents.

Variable	Descriptive
Age, median (1 Q; 3 Q)	29 (26.3; 30.8)
Weeks in the oncology service, per year, median (1 Q; 3 Q)	6 (4; 38)
Sex, n (%) ^a^	
Female	21 (75)
Male	7 (25)
Ethnicity, n (%) ^a^	
White	17 (60.7)
Black	1 (3.6)
Asian	1 (3.6)
Other	9 (32.1)
Religion, n (%) ^a^	
Buddhist	1 (3.6)
Catholic	15 (53.6)
Spiritist	1 (3.6)
Evangelical—Other Christian	4 (14.3)
Agnostic	7 (25)
Year of medical residency, n (%)	
1	7 (24.1)
2	6 (20.7)
3	8 (27.6)
4	8 (27.6)
Region of medical residency, n (%)	
Belém-Pará	20 (69.1)
Recife	4 (13.7)
São Paulo	5 (17.2)
Internship in palliative care, n (%)	
No	4 (13.8)
Yes	25 (86.2)
Medical postgraduate training, n (%)	
Internal Medicine	27 (93.1)
Geriatrics	5 (17.2)
Hematology	6 (20.7)
Palliative medicine	2 (6.9)
Clinical Oncology	2 (6.9)

^a^ One participant did not answer this question.

**Table 2 ijerph-20-03239-t002:** Barriers according to years of medical residency.

	Up to 2 Years (R1 + R2) Med (1 Q; 3 Q)	More than 2 Years (R3 + R4) Med (1 Q; 3 Q)	Mann–Whitney’s *p*-Value
(A.1) Barriers Related to the Patient/Family			
Patient does not have an advance directive	5 (3; 7)	6 (5.3; 7)	0.189
Patients’ difficulty in understanding their prognosis/length of survival	6 (5; 7)	7 (6; 7)	0.158
(A.2) Barriers Related to the Physician			
Uncertainty in estimating prognosis/length of survival	5 (3.5; 5.5)	5 (5; 6)	0.202
Lack of training to have these conversations (e.g., not sure about what to say, etc.)	5 (3; 6.5)	6 (4.3; 7)	0.138
(A.3) Barriers Related to the System or External Factors			
Lack of time to have conversations with patients/family	4 (2.5; 5.5)	6 (4.3; 7)	0.034 *
Brief therapeutic relationship with patient/family (e.g., patient newly referred to you)	5 (3.5; 6)	6 (5; 6.8)	0.073
(B) Barriers to Discontinuation of Cancer-Directed Therapies(B.1) Barriers Related to the Physician			
You are unsure of the benefits of further active cancer therapy in this patient	5 (3.5; 5.5)	5 (4.3; 6)	0.297
Difficulty estimating patient prognosis/length of survival	4 (3; 5.5)	5 (3; 6)	0.398
Patient age (for example, difficulty talking about it with younger people)	3 (2; 5.5)	5 (5; 6)	0.025 *
(B.2) Patient and Family Barriers			
Patient’s poor appreciation or denial of likely survival duration	6 (5; 6)	6 (6; 7)	0.136
Patient’s inflated expectation of the benefits from further cancer-directed therapy	5 (3; 6)	7 (6; 7)	<0.001 *
(B.3) External Barriers			
Lack of decision aids for discontinuing cancer-directed therapy	5 (3.5; 5)	5 (4.3; 6.8)	0.160
Lack of guidelines for discontinuing cancer-directed therapy	5 (3; 6)	5 (3; 6)	0.768
(C) Timing of Palliative Care Referral			
Lack of access to palliative care services within the hospital	6 (4.5; 7)	6 (5; 7)	0.566
Lack of sufficient allied health support (social workers, nurse practitioners, etc.) to aid in patient support/referral process	6 (5; 7)	6 (5; 7)	0.838
(D) Willingness to Participate in Communication and Decision-Making About Goals of Care			
Rate your willingness to initiate the discussion (bring up the subject) about goals of care with patients such as these and their families	6 (5; 7)	6 (6; 7)	0.305
Rate your willingness to lead the discussion with patients such as these and their families. This includes exchanging information (disclosing diagnosis, prognosis, and eliciting values) and being a decision coach (clarifying values, assisting with weighing options for care, etc.)	6 (5.5; 6.5)	6 (6; 7)	0.230

* for significant differences (*p* < 0.05); Med—median.

**Table 3 ijerph-20-03239-t003:** Training on communication skills.

Self-Reported Level of Skill on Having Goals of Care Decisions n (%)	
Limited	0 (0)
Fair	3 (10.3)
Average	13 (44.8)
Very good	12 (41.4)
Expert	1 (3.4)
Priority (from 1 to 5) for learning the communication skill, n (%)	
3 (Moderately Priority)	3 (10.3)
4 (Moderately High Priority)	13 (44.8)
5 (High Priority)	13 (44.8)
Formal training regarding communication with patients andfamilies about goals of care? n (%)	
No	2 (6.9)
Yes	27 (93.1)
If yes, the quality of training, n (%)	
Moderately low	1 (3.7)
Neither high nor low	5 (18.5)
Moderately high	9 (33.3)
Very high	6 (22.2)
Extremely high	6 (22.2)
The importance of spirituality or religion in your life ^a^, n (%)	
Extremely unimportant	2 (7.1)
Very unimportant	1 (3.6)
Somewhat unimportant	1 (3.6)
Somewhat important	7 (25)
Very important	7 (25)
Extremely important	10 (35.7)

^a^ One participant did not answer the question.

**Table 4 ijerph-20-03239-t004:** Suggestions to facilitate decision-making about goals of care in clinical practice (n = 27).

Strategies Related to the Physician (Intrinsic Factors)	n (%)
Communication skills training	09 (33)
Physician–patient relationship	07 (25)
Early conversation about prognosis	06 (22)
Empathetic communication	05 (18)
Strategies related to the system (extrinsic factors)	n (%)
Availability to discuss goals of care	04 (14)
Access to palliative care and interdisciplinary teams	03 (11)

## Data Availability

The evaluation questionnaire of this research was adapted to Portuguese language from the study of Ethier J-L et al. Perceived Barriers to Goals of Care Discussions With Patients With Advanced Cancer and their Families in the Ambulatory Setting: A Multicenter Survey of Oncologists. *J. Palliat. Care*
**2018***, 33*, 125–142. The Portuguese version is in the link: https://drive.google.com/file/d/1TR2VgzA74VCVdR38wMmr2FmrNZQMSabv/view?usp=share_link.

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
