# Peer review of "Advance Care Planning and Goals of Care Discussion: Barriers from the Perspective of Medical Residents"

_ijerph, 2023, doi:10.3390/ijerph20043239_

Round 1
Reviewer 1 Report (Previous Reviewer 1)
The authors have improved the manuscript.
Reviewer 2 Report (Previous Reviewer 4)
The authors have substantially improved the manuscript
This manuscript is a resubmission of an earlier submission. The following is a list of the peer review reports and author responses from that submission.
Round 1
Reviewer 1 Report
I am thankful for giving me the chance to review the manuscript entitled “Advance Care Planning And Goals of Care Discussion: Barriers From The Perspective Of Brazilian Medical Residents”. The topic is with novelty and the clinical significance is existed. The manuscript is well designed and written. I suggest to describe about for patients with cancer or in clinical oncology in the title.
Author Response
Thank you for the sugestion. We improved the introduction, methods and results, better characterizing the population studied.
Reviewer 2 Report
Reviewer comments
A. Strength of study:
a. The clinical relevance of the topic re: importance of quality, empathetic and informed goals of care and possibly end of life discussions with patients experiencing life-threatening conditions, need palliative care and must make difficult decisions with their loved ones.
b. The study valued the perspectives of medical residents who are critical in contributing to the topic and future practice.
c. A multicenter study
d. Relevance of having medical residents as research participants: future viable practitioners to plan effective future programs for clinicians.
e. Rigor of validating and language translation of the tool/questionnaire. Discussed goals of care in “Brazilian reality” and language.
B: Reviewer Recommendations:
1. Aim of the study/Results: Line 61-62: “To explore barriers to goals of care and advance care planning discussion in patients with cancer from the perspective of Brazilian medical residents”.
a. Consistent with the results of the study, it appears you did some comparison analysis based on the number of years in the residency program. It may be a great idea mention that in the purpose of the study
b. Lines 173-184: “Suggestions for improving communication and decision-making about goals of care”. Some quotes from the participants were provided. Please mention if this study has a qualitative component. Are the questions necessitating these responses in quotes part of the validated toll/questionnaire used to collect data?
2. Line 308: “Institutional Review Board Statement: Ethical approval was waived by the University Hospital from our region”. What was the waiver based on? Were the medical residents’ part of a previous study, and you are just analyzing the data now?
3. Figure 1: ineligible figures for readers
Figure 1. Barriers of patient, physician and external factors to initiating goals of care discussions (A), to interrupt cancer- 159 directed therapies (B), and to refer to palliative care (C) (median scores).
4. Discussion section.
We understand that the authors did not have the statistical power for inferential statistics like regression analysis. The Mann Whitneys U non-parametric testing was conducted. With the final sample size of 29 from 94 distributed survey, there is possible sample bias.
Please further discuss limitation of possible sample bias and its impact on the interpretation of findings, especially since the ten of the participants who completed the paper surveys were all invited participants before an educational program.
Author Response
Dear reviewer,
Thank you for the suggestions. The responses follow below.
- Aim of the study/Results: Line 61-62: “To explore barriers to goals of care and advance care planning discussion in patients with cancer from the perspective of Brazilian medical residents”.
a) Consistent with the results of the study, it appears you did some comparison analysis based on the number of years in the residency program. It may be a great idea mention that in the purpose of the study
R- we adjusted the aim, including this topic. "To explore barriers to goals of care and advance care planning discussion in patients with cancer from the perspective of Brazilian medical residents, and to compare such barriers according to years of medical residency."
b) Lines 173-184: “Suggestions for improving communication and decision-making about goals of care”. Some quotes from the participants were provided. Please mention if this study has a qualitative component. Are the questions necessitating these responses in quotes part of the validated toll/questionnaire used to collect data?
R- Thank you. We took them out of the quotes.
2. Line 308: “Institutional Review Board Statement: Ethical approval was waived by the University Hospital from our region”. What was the waiver based on? Were the medical residents’ part of a previous study, and you are just analyzing the data now?
R- Thak you for the advice. We adjusted this sentence with ethics approval. "This study was submitted and approved by the Ethics Committee of João de Barros Barreto University Hospital (No 4.376.747). All subjects had to give their informed consent to participate in this study."
3. Figure 1: ineligible figures for readers
R- Thank you. We improved the quality of the figure.
- Discussion section.
We understand that the authors did not have the statistical power for inferential statistics like regression analysis. The Mann Whitneys U non-parametric testing was conducted. With the final sample size of 29 from 94 distributed survey, there is possible sample bias.
R- We included a sentence in discussion about this topic.
"Nonetheless some limitations included a small sample, because it involved only medical residents from three university hospitals. Furthermore, a positive response bias among those who did participate is possible, with potential for higher response rates among those most engaged in palliative and end-of-life care."
Reviewer 3 Report
The manuscript “Advance Care Planning And Goals of Care Discussion: Barriers From The Perspective Of Brazilian Medical Residents” has been reviewed. This topic is important as the world is now facing challenge of increasingly elderly population and we do need young medical doctors to be able to discuss with elderly patients about life-threatening disease. However, the article has several issues that need to be addressed:
1. Why this research topic is important in Brazil? This should be addressed with more context of culture of Brazil or South America. For instance, past researches as follows demonstrated that ACP concept is not widely accepted in Brazil, but is important to reduce futile treatment among cancer patients. Research like this could help readers to understand why this issue is important. The introduction section should be strengthened.
Hassegawa LCU, Rubira MC, Vieira SM, Rubira APA, Katsuragawa TH, Gallo JH, Nunes RML. Approaches and reflexions on advance healthcare directives in Brazil. Rev Bras Enferm. 2019 Jan-Feb;72(1):256-264.
de Wylson Fernandes Gomes de Mattos D, Thuler LC, da Silva Lima FF, de Camargo B, Ferman S. The do-not-resuscitate-like (DNRL) order, a medical directive for limiting life-sustaining treatment in the end-of-life care of children with cancer: experience of major cancer center in Brazil. Support Care Cancer. 2022 May;30(5):4283-4289.
2. Same, why medical residents? There should be some words discussing about the importance of exploration attitude of medical residents on ACP and past studies in the Introduction. Such as:
Deep KS, Green SF, Griffith CH, Wilson JF. Medical residents' perspectives on discussions of advanced directives: can prior experience affect how they approach patients? J Palliat Med. 2007 Jun;10(3):712-20. doi: 10.1089/jpm.2006.0220. PMID: 17592983.
3. Of 94 residents, only 29 responded. Selection bias could happen. To lower possible selection bias, do authors have socio-demographic profile of those who did not respond?
4. There were 2 kinds of instruments used: paper questionnaire and electronic questionnaire. And the response rate was largely different. Selection bias could also happen, please discuss it in the Discussion section.
5. One participant did not answer many questions. Why included the participant?
6. For Figure 1, the quality of the figure is low, please enhance it.
7. Is line 173 a heading? It looked like a heading.
8. The conclusion should be more stronger. From the results, what should we do to improve the communication skills and positive attitude toward ACP? The authors mentioned some in the Discussion, please add them into Conclusion section to make the manuscript with more impact.
Author Response
Dear Reviewer, Thank you for all considerations.
- Why this research topic is important in Brazil? This should be addressed with more context of culture of Brazil or South America. For instance, past researches as follows demonstrated that ACP concept is not widely accepted in Brazil, but is important to reduce futile treatment among cancer patients. Research like this could help readers to understand why this issue is important. The introduction section should be streghtened
- Same, why medical residents? There should be some words discussing about the importance of exploration attitude of medical residents on ACP and past studies in the Introduction. Such as:
R-1-2- Thank you for the suggestions. We includding some references suggested to reinforce the importance of these topics ( need to discuss this topic with Brazilian residents, since the scarcity of studies including this population).
- Of 94 residents, only 29 responded. Selection bias could happen. To lower possible selection bias, do authors have socio-demographic profile of those who did not respond?
- There were 2 kinds of instruments used: paper questionnaire and electronic questionnaire. And the response rate was largely different. Selection bias could also happen, please discuss it in the Discussion section.
R- 3-4: We adjusted the methods and the discussion, including the discussion about selection bias.
- One participant did not answer many questions. Why included the participant?
R- He did not answer only 3 questions of the sociodemographic questionnaire (sex, religion and ethnicity), and a question about the importance of spirituality. But he answered most questions (all about barriers to discussing goals of care)
- For Figure 1, the quality of the figure is low, please enhance it.
Thank you for advice, we enhanced the quality of figure.
- Is line 173 a heading? It looked like a heading.
Yes, we adjusted.
- The conclusion should be more stronger. From the results, what should we do to improve the communication skills and positive attitude toward ACP? The authors mentioned some in the Discussion, please add them into Conclusion section to make the manuscript with more impact.
R- Thank you, we adjusted the conclusion.
.
Reviewer 4 Report
LINE 84-92: The authors should explain how they have carried out the questionnaire, whether with experts, ad hoc, with piloting....
LINE 97-98: “This study was submitted and approved by the Ethics Committee. All participants provided written informed consent” …… one the identification number granted by the committee or by the committee or where it has been authorized.
LINE 104-105: Out of ninety-four resident 104 physicians referred, 29 responded. It is an excessively low sample. Why did 67% of the subjects under study not answer?
RESULTS IN GENERAL: The results can be more attractive and attractive if they are displayed in diagrams or with histograms and correlating data.
REFENCES AND CITES: I do not see in the text the bibliographic reference number 19 and 20.
IN GENERAL: In studies where the results of interventions, or their results, are measured, the sample may be smaller; however, I consider that in questionnaires, in order for it to be published with a certain scientific rigor, the number of participants should be increased in order to demonstrate the good work done.
Author Response
Dear reviewer, Thank you for the suggestions,
LINE 84-92: The authors should explain how they have carried out the questionnaire, whether with experts, ad hoc, with piloting....
R-In methods we explain how the process of cross-cultural adaptation and translation of the questionnaire used in the research was carried out.
LINE 97-98: “This study was submitted and approved by the Ethics Committee. All participants provided written informed consent” …… one the identification number granted by the committee or by the committee or where it has been authorized.
R- Thank you for advice. We adjusted this paragraph.
"This study was submitted and approved by the Ethics Committee of João de Barros Barreto University Hospital (No 4.376.747). All subjects had to give their informed consent to participate in this study."
LINE 104-105: Out of ninety-four resident 104 physicians referred, 29 responded. It is an excessively low sample. Why did 67% of the subjects under study not answer?
R- There were 29 responses from a total of 94 residents. We adjust in methods. Probably because of the pandemic - the survey was carried out in 2021, many health professionals are overburden, besides, the nature of the topic, not always easy to discuss. In discussion we improved this topic, in limitations of the study.
WE improved the figure 1, with the results.
REFENCES AND CITES: I do not see in the text the bibliographic reference number 19 and 20.
"The use of decision aids and advanced technology, such as audio-visual material and online tutorials; workshops using role play, training content focused on ACP communication skills and the needs and experience of patients in decision-make processes, contributing to ACP training programs effectiveness19,20." On page 15, discussion.